# The Moderator Role of Interpersonal Emotion Regulation on the Associations between Commitment, Intimacy, and Couple Satisfaction

**DOI:** 10.3390/ijerph191710506

**Published:** 2022-08-23

**Authors:** Mihaela Jitaru, Maria Nicoleta Turliuc

**Affiliations:** Faculty of Psychology and Educational Sciences, Alexandru Ioan Cuza University of Iasi, 700554 Iasi, Romania

**Keywords:** interpersonal emotion regulation, commitment, intimacy, couple satisfaction, APIMoM, dyadic analysis

## Abstract

Couple satisfaction is seen as very important by all those in a romantic relationship; however, there are no recipes for it. Using a dyadic approach, we investigate how commitment and intimacy influence couple satisfaction and the moderator role of interpersonal emotion regulation (affect-improving and -worsening strategies). To achieve the scope of the study, we collected data from 131 couples, which were later analyzed using the actor–partner interdependence model with moderation (APIMoM). The results showed that the actor-effect of both commitment and intimacy on couple satisfaction is significant. We found mixed results for the partner-effect of the two variables. Both partners’ strategies moderated the association between commitment and couple satisfaction. Women’s use of affect-worsening strategies moderated the link between men’s intimacy and women’s couple satisfaction. The impact of the interactions of commitment or intimacy with interpersonal affect-improving and -worsening strategies on couple satisfaction is discussed further, as well as the implications and importance of the results.

## 1. Introduction

Romantic relationships play an essential role in everyone’s life; therefore, the need to have them be healthy, strong, stable, and durable is understandable. However, not everyone can have such a relationship. An indicator often used to assess how the relationship is going is the general couple satisfaction reported by the partners. Couple satisfaction is a subjective global evaluation of the relationship made by one or both partners [1] or the positive evaluation and attraction to the relationship [2,3]. The previous studies highlighted that a higher level of couple satisfaction was associated with the relationship’s longevity [4], with partners’ overall health [5], satisfaction with life [6], and well-being [7]. Considering all these favorable outcomes, highlighting and researching those factors that can contribute to a higher level of couple satisfaction are essential endeavors. The factors influencing couple satisfaction were intimacy, commitment [8,9,10], emotion regulation, and many others. This study aimed to verify the moderating influence of interpersonal emotion regulation (IER) on the relation between intimacy or commitment and couple satisfaction.

### 1.1. Commitment and Couple Satisfaction

Relational commitment is commonly understood as an individual’s intention or desire to continue their relationship [11,12]. It is also characterized by a long-term orientation and a strong attachment to the partner and the relationship [13,14]. Therefore, it is often viewed as a cognitive activity that impacts the decisions one is making about their relationship and partner [3,15].

In multiple instances, it was revealed that relational commitment and couple satisfaction hold a strong association [10,16,17,18]. The investment model [2,19] aimed to explain the connections between commitment and couple satisfaction. It explored why some relationships persist in time while others wither and dissolve. It theorized that satisfying experiences will not suffice for a long-term relationship. Other factors, like social pressure to continue, complicated termination procedures, and availability of alternative romantic partners, also had a significant impact on the relationship [20].

The investment model of commitment proposed that satisfaction can be a predictor of commitment; yet, some authors suggested that the inverse relation should be further explored [8,9]. Previous research suggested that high levels of commitment were associated with high levels of marital satisfaction and that commitment was one of the most critical factors for a good relationship [21]. Furthermore, it showed that low levels of commitment often revealed that the partner offered less and obtained only minor benefits from the relationship and thus had low levels of satisfaction [22].

### 1.2. Intimacy and Couple Satisfaction

Romantic intimacy was defined as a process involving the feeling of closeness to the partner and the wish to share experiences and activities with them [23,24]. It is relevant to highlight the interactional aspect of intimacy, as it is not unidirectional. The action produced by one partner will elicit a responsive reaction from the other [24]. Individuals with higher levels of intimacy tend to be more open, supportive, understanding, caring, and honest in their romantic relationships [25]. In previous research, intimacy emerged as an essential factor in couple satisfaction, as those with a higher level of intimacy had a more satisfactory relationship [26]. Similar to couple satisfaction, intimacy can positively affect both the individual and the relationship. Intimacy was revealed to positively impact physical health [27], well-being [28,29], and couple satisfaction [30,31,32].

### 1.3. Interpersonal Emotion Regulation and Romantic Relationships

Interpersonal emotion regulation (IER) is a process through which an individual (regulator) aims to change and influence another’s (target) emotional state [33,34,35]. Niven, Totterdell, and Holman’s model distinguished two types of regulating strategies, the interpersonal affect-improving ones and the interpersonal affect-worsening ones. The strategies that improve the affect represent all the behaviors and actions that a regulator employs to elicit, raise the level of positive emotions or lower the level of negative emotions in their target [33,34]. The affect-worsening strategies encompass all the behaviors and actions that a regulator employs intending to elicit negative emotions, lower the level of positive emotions, or raise the level of negative emotions in their target [33,34].

IER, due to its very nature, had an effect on social relationships and, therefore, on romantic relationships [36]. Using IER (such as touch) in romantic dyads helped promote the psychological well-being of the partners [37]. Through IER (down-regulation of negative emotions), partners reduced in each other the level of distress they were experiencing [38]. In turn, this helped maintain a high level of couple satisfaction [39]. Couple satisfaction was also directly and positively influenced by IER (enhancing positive affect) [40]. As previously seen, IER strategies benefited couples’ satisfaction; yet, they needed to be from the affect-improving category, as seen in the previous examples. In opposition, when using strategies meant to worsen the affect, a negative impact on couple satisfaction was most likely to be seen. Hostile criticism [41], ridicule [42], withdrawal, and punitive intent [43] had negative connections to couple satisfaction.

### 1.4. Commitment and Interpersonal Emotion Regulation

Both commitment and IER are processes specific to interpersonal relationships. Romantic commitment was described as an individual’s desire to continue their relationship and having a strong attachment to both their partner and the relationship [13,14]. IER represents the actions taken by a person to influence another individual’s affective state, in this case, their romantic partner [33,34]. Commitment and interpersonal affect-worsening strategies (withdrawal behaviors, criticism, belittling, disrespect) were negatively connected [44,45,46,47]. In opposition, interpersonal affect-improving strategies (support, allowing the partner to present their emotions) were positively related to commitment [47,48].

### 1.5. Intimacy and Interpersonal Emotion Regulation

As stated previously, intimacy is an interactive process between the individuals in a romantic relationship [24,49]; thus, making it an interpersonal process. Due to its characteristics, IER is also a relational process [33,34]. Intimacy, more specifically, disclosure, has been established as an indirect socio-affective pathway to emotion regulation [37,50]. In addition, the relationships that one has throughout their life have been highlighted as essential resources in emotional regulation [51,52,53,54,55].

### 1.6. The Present Study

This research aimed to investigate the relationship between the main study’s variables and, most importantly, show how commitment and intimacy impact romantic relationships and interpersonal emotion regulation moderates those relationships. In addition, all these connections are explored from a dyadic perspective using the Actor–Partner Interdependence Model [56]. This model allows researchers to investigate the associations between an individual’s traits and their outcomes (actor-effect) and partner outcomes (partner-effect). Previous research showed that commitment and intimacy are among the factors that impact the most couple satisfaction [21,26]. IER is a relatively new concept that needs investigation for a better understanding, especially concerning other relationship-relevant factors. However, some behaviors that are part of IER strategies were connected to romantic commitment [44,48], intimacy [50], and couple satisfaction [40,43]. Therefore, this study explored the relations between commitment/intimacy and couple satisfaction with the moderating role of IER in both its forms (affect-improving strategies and affect-worsening strategies). Given what we know till this moment of the connections between previously stated variables, we expected that both commitment and intimacy would have a positive effect (actor and partner) on couple satisfaction. In addition, we expected a positive impact of the use of affect-improving strategies on the relationship between commitment/intimacy and couple satisfaction and a negative one for the use of affect-worsening strategies.

## 2. Materials and Methods

### 2.1. Participants

This study’s data were gathered from 131 heterosexual young couples (N = 262). The inclusion criteria for this study were the following: the participants had to be at least 18 years old and in a romantic relationship for at least six months at the moment they took the survey, and both of the couple partners had to complete all of the questionnaires. The mean age for women was 21.76 years (SD = 3.13), and for men it was 23.60 years (SD = 3.58), and the average relationship length was 26.35 months (SD = 26.68). Of the total of 131 couples, 43 couples were living together, while the rest of the couples were not living in the same household when they took the survey. All the participants gave their informed consent to participate in the study and were informed about the anonymity of their answers and the fact that they could withdraw from the study without any consequences. The respondents’ participation was voluntary, and they were not compensated.

### 2.2. Measures

The Couple Satisfaction Index−4 (CSI−4) [57] was used to assess the general *couple satisfaction*. CSI−4 is a four-item scale developed using the item response theory. The total score was calculated by summing the answers given by the respondents to the items (e.g., ‘I have a warm and comfortable relationship with my partner’). The internal consistency in this study’s sample was high, α = 0.90 for women and α = 0.91 for men.

The *commitment* was assessed with the commitment subscale from the Investment Model Scale (IMS) [58]. This subscale comprises seven items (e.g., ‘I am committed to maintaining my relationship with my partner.’). For each item, the respondents reported their degree of agreement on a scale ranging from 0 (do not agree at all) to 8 (agree completely). The total score was calculated by summing the answers given to each item. The internal consistency coefficients were high both for women and men (αw = 0.88, αm = 0.93).

The Personal Assessment of Intimacy in Relationships (PAIR) [23] was used to assess the respondents’ *couple intimacy*. The scale consisted of 36 items (e.g., ‘My partner listens to me when I need someone to talk to.’) that measure the self-reported feelings of intimacy along five dimensions (emotional, social, sexual, intellectual, and conventionality). Because the subscales are highly correlated they can be combined to form a composite measure of intimacy [59]. The internal consistency was high both for women (α = 0.85), and men (α = 0.83).

The *interpersonal emotion regulation* was assessed using the Emotion Regulation of Others and Self (EROS) [60]. For the purpose of this study, only nine items were used, the ones that appertain to the extrinsic subscale. Furthermore, they were modified to fit the aims of this study better. Therefore, we changed the generalist “someone” to the more focused “my partner” in each item. Of the nine items, three of them were for assessing the affect-worsening strategies (e.g., ‘I told my partner about their shortcomings to try to make them feel worse.’), while the other six items assessed the use of affect-improving strategies (e.g., ‘I did something nice with my partner to try to make them feel better.’). The total score for the two subscales was calculated by summing the answers given by the participants for each item on a scale ranging from 1 (not at all) to 5 (a great deal). The internal consistencies for the subscales were acceptable for this research (affect-worsening strategies: α = 0.68 for women, α = 0.67 for men; affect-improving strategies: α = 0.80 for women, α = 0.77 for men).

Additionally, the *demographic variables* we assessed were age, relationship length, and the living situation (whether the couple lived together or in different households). All the items are presented in Appendix A.

### 2.3. Statistical Analysis

Firstly, we computed paired-sample *t*-tests to assess the gender differences for the main variables and partial correlations, controlling for age, relationship length, and the living situation (together or in separate households). Secondly, using the Actor–Partner Moderation Model (APIMoM) [61], we computed four different analyses (as observed in the conceptual model from Figure 1). The first analyzed the relationship between commitment and couple satisfaction, moderated by the use of interpersonal affect-improving strategies, while the second changed the moderator to the use of interpersonal affect-worsening strategies. The third model investigated the relationship between intimacy and couple satisfaction, moderated by the interpersonal affect-improving strategies, and the fourth model kept the main variables, changing the moderator to the interpersonal affect-worsening strategies. After analyzing the four models, their terms were graphed following Dawson’s [62] indications for a more straightforward interpretation of the significant interactions. The statistical analyses were computed using IBM SPSS version 20 (SPSS, Inc., Chicago, IL, USA) and IBM SPSS Amos version 22 (SPSS, Inc., Chicago, IL, USA).

## 3. Results

Firstly, we present the preliminary analyses, which comprise the results of the paired-sample *t*-tests for the gender differences for the main variables and the correlations for the same variables. Secondly, we present the results separately for the four APIMoM models we computed. For each model, we present the fit indices, highlight the significant results, and for the significant interaction effects, we provide the interaction plots. The first model contains the commitment of both partners as predictors, their couple satisfaction separately as outcomes, and their use of interpersonal affect-improving strategies as moderators. The second model follows the precedent’s structure, changing only the moderators, here being the interpersonal affect-worsening strategies. In the third and fourth models, we have the level of intimacy of both partners as predictors and their couple satisfaction, separately, as outcomes. In the third model, the moderator is the interpersonal affect-improving strategies, and the affect-worsening strategies is the moderator in the fourth model. Before starting to present the results of this study, we need to clarify that the couples that completed the surveys were young, with an average age of 23 years old. This means that the results of this study cannot be reliably generalized to older individuals that are in a romantic relationship.

### 3.1. Preliminary Analysis

Firstly, we conducted a series of paired-sample *t*-tests to assess the gender differences for the main variables of the study. The results (Table 1) show significant differences between women and men only for the use of interpersonal affect-worsening strategies t (130) = 2.60, *p* = 0.011. Table 2 displays the correlational matrix for the study’s main variables. Table 2 shows the correlational matrix for the study’s main variables. Both women’s and men’s couple satisfaction were positively and significantly associated with each other and with women’s and men’s commitment, intimacy, and use of affect-improving strategies and negatively associated with the use of interpersonal affect-worsening strategies.

### 3.2. The Effect of Romantic Commitment on Couple Satisfaction Moderated by the Use of Affect-Improving Strategies

This model showed very good fit indices: χ²(3) = 3.502, *p* = 0.321, χ²/df = 1.167, CFI = 0.999, RMSEA = 0.036 (0.000–0.156). The results (Table 3) show that women’s and men’s romantic commitment had a positive and significant effect on their couple satisfaction (βwoman = 0.234, βman = 0.265). The partner effects of commitment on couple satisfaction were not significant for either women or men (βwoman = 0.043, βman = 0.070). The women’s use of interpersonal affect-improving strategies significantly affected their couple satisfaction (βwoman = 0.197) but not their partners’ couple satisfaction (βman = 0.028). Men’s use of interpersonal affect-improving strategies had no significant effect on either men’s or women’s couple satisfaction.

The interaction effect between women’s romantic commitment and women’s use of affect-improving strategies was significant for both women’s and men’s couple satisfaction (βwoman = 0.028, βman = 0.019). The interaction plot (Figure 2a) suggests that the slopes for the relationships between women’s commitment and women’s satisfaction were positive for both low and high women’s use of interpersonal affect-improving strategies. However, it can be observed that for the women who used more interpersonal affect-improving strategies and were more committed to their relationships, the effect is more accentuated.

In the second interaction plot (Figure 2b), we noticed that the relationship between women’s commitment level and men’s couple satisfaction varied, but only when women used more interpersonal affect-improving strategies. Therefore, when women had a higher level of commitment and used more interpersonal affect-improving strategies, their partners had a higher couple satisfaction.

The interaction effects between men’s commitment and women’s use of affect-improving strategies were significant for both women’s and men’s couple satisfaction (βwoman = −0.018, βman = −0.023). For the first effect, the interaction plot (Figure 2c) suggests that women’s couple satisfaction did not noticeably vary based on their partners’ level of commitment when the women used high levels of interpersonal affect-improving strategies. On the opposite side, when the women used less interpersonal affect-improving strategies, there was a positive relationship between their partners’ commitment and their own satisfaction.

In the next interaction plot (Figure 2d), we can observe that both slopes indicate positive relations. However, it is important to note that men’s couple satisfaction seems higher when they had a higher commitment and their partners used less interpersonal affect-improving strategies.

The interaction effects between women’s commitment and men’s use of interpersonal affect-improving strategies were significant for men’s couple satisfaction but not for women’s couple satisfaction (βwoman = 0.010, βman = −0.027). The interaction plot (Figure 3a) computed for the significant effect indicates that there was a positive relationship between women’s commitment and men’s satisfaction only when men used less affect-improving strategies.

The interaction effects between men’s romantic commitment and men’s use of interpersonal affect-improving strategies had a significant impact only on men’s couple satisfaction (βwoman = −0.015, βman = 0.039). The computed interaction plot (Figure 3b) suggests that men’s couple satisfaction was the highest when they were committed to their romantic relationship and used more interpersonal affect-improving strategies.

### 3.3. The Effect of Romantic Commitment on Couple Satisfaction Moderated by the Use of Affect-Worsening Strategies

The model revealed very good fit indices: χ²(1) = 0, *p* = 0.991, χ²/df = 0, CFI = 1, RMSEA = 0 (0.000–0.000). The results (Table 3) indicate that women’s and men’s romantic commitment significantly and positively affected their couple satisfaction (βwoman = 0.229, βman = 0.170). The partner effect of commitment on couple satisfaction was significant and positive for both women and men (βwoman = 0.066, βman = 0.100). In the results, we can also observe that women’s use of interpersonal affect-worsening strategies did not significantly influence their couple satisfaction (βwoman = −0.123); however, it affected their partners’ satisfaction in a significant and negative way (βman = −0.211). Men’s use of interpersonal affect-worsening strategies had no significant impact on their or their partners’ couple satisfaction (βwoman = −0.099, βman = −0.078).

The results do not show a significant interaction effect of women’s commitment and use of interpersonal affect-worsening strategies on women’s or men’s couple satisfaction (βwoman = −0.012, βman = −0.017). In comparison, the interaction of men’s commitment and women’s use of interpersonal affect-worsening strategies had a significant and positive effect on both women’s and men’s couple satisfaction (βwoman = 0.030, βman = 0.030). The interaction plot (Figure 4a) for women’s couple satisfaction indicates that there was a positive relationship between men’s commitment and women’s satisfaction only when women used more affect-worsening strategies.

In the interaction plot for men’s couple satisfaction (Figure 4b), we can observe that there was a positive relationship between men’s commitment and their satisfaction regardless of the women’s level of affect-worsening strategies. However, the relationship was stronger when women used more affect-worsening strategies. For this model, none of the other interactions significantly affected women’s or men’s couple satisfaction (Table 3).

### 3.4. The Effect of Intimacy on Couple Satisfaction Moderated by the Use of Affect-Improving Strategies

The analyzed model showed very good fit indices: χ²(1) = 0.025, *p* = 0.875, χ²/df = 0.025, CFI = 1, RMSEA = 0 (0.000–0.119). The results (Table 4) display a positive and significant effect of intimacy on couple satisfaction for both women and men (βwoman = 0.084, βman = 0.103). The partner effect was significant only for men (βman = 0.047); women’s level of intimacy positively influenced their couple satisfaction. In comparison, women’s couple satisfaction was not affected by men’s relational intimacy (βwoman = 0.024). Women’s use of interpersonal affect-improving strategies had a positive and significant effect on their couple satisfaction (βwoman = 0.180) but no significant impact on their partner’s couple satisfaction (βman = 0.030). Men’s employment of interpersonal affect-improving strategies had no impact on their own or their partner’s couple satisfaction (βwoman = 0.042, βman = 0.082). As can be observed in Table 4, there were no significant interaction effects for this model.

### 3.5. The Effect of Intimacy on Couple Satisfaction Moderated by the Use of Affect-Worsening Strategies

The model showed a very good fit: χ²(1) = 0.129, *p* = 0.720, χ²/df = 0.129, CFI = 1, RMSEA = 0 (0.000–0.166). This model’s results (Table 4) indicate that intimacy had a positive and significant effect on couple satisfaction for women and men (βwoman = 0.091, βman = 0.104). The partner effect suggests that only men’s couple satisfaction was significantly affected by women’s level of intimacy (βwoman = 0.023, βman = 0.040). In addition, neither women’s (βwoman = −0.043, βman = −0.091) nor men’s (βwoman = −0.084, βman = −0.067) use of interpersonal affect-worsening strategies had a significant effect on women’s or men’s couple satisfaction. The interaction effects of women’s intimacy and their use of interpersonal affect-worsening strategies on their or their partners’ couple satisfaction were not significant (βwoman = 0.002, βman = −0.001). The interaction of men’s level of intimacy with women’s use of interpersonal affect-worsening strategies had a significant effect on women’s couple satisfaction (βwoman = 0.015) but not on men’s satisfaction (βman = 0.002).

The interaction plot (Figure 4c) for the significant effect shows that men’s intimacy and women’s satisfaction were positively associated only when women used more affect-worsening strategies. The interaction between women’s intimacy and men’s use of interpersonal affect-worsening strategies did not significantly affect women’s or men’s couple satisfaction (βwoman = −0.013, βman = −0.006). Similarly, the interaction of men’s intimacy and their use of interpersonal affect-worsening strategies did not significantly affect women’s or men’s couple satisfaction (βwoman = 0.005, βman = 0.006).

## 4. Discussion

Couple satisfaction is a characteristic most people in a romantic relationship desire. However, how to reach it is not always clear and there are no general recipes. That makes exploring the possible mechanisms that contribute to the changes in couple satisfaction all the more desirable. Consequently, this research explored the effects of commitment and intimacy on couple satisfaction while adding the moderating effect of IER. This study and its results focused on young couples; therefore, the following explanations and implications need to be understood through this lens.

Prior to investigating the effects mentioned earlier, we verified if there were gender differences in our variables’ scores due to the dyadic character of our investigation process. The only differences observed were for the use of interpersonal affect-worsening strategies. Prior research did not show a clear pattern of connections between gender and the use of interpersonal affect-improving and -worsening strategies. Some studies showed differences between women’s and men’s use of interpersonal affect-worsening strategies, with the latter employing more [60,63]. In contrast, other studies did not find gender differences [64,65]. However, there is a study that obtained similar results, with women using more interpersonal affect-worsening strategies than men [66]. Given these contradictory results, more investigation of this process is necessary.

We observed that the actor-effect of commitment on couple satisfaction was a significant one. These results align with previous studies that showed similar effects [9,21,22]. The partner-effect of commitment on couple satisfaction was significant only in the model that included the use of interpersonal affect-worsening strategies as a moderator, which means that the men’s level of commitment positively affected their partners’ couple satisfaction. Women’s commitment influenced men’s couple satisfaction similarly. Therefore, commitment contributes significantly to couple satisfaction [21]. The factors contributing to commitment, compared to alternatives or investment size [2,19], can explain why it impacts couple satisfaction. Therefore, if an individual, after evaluating the possible alternatives, concludes that their relationship and their partner are the best possible for them, it will increase their couple satisfaction. Furthermore, it is possible, as the present results seem to indicate, that seeing one’s partner’s high level of commitment increases their couple satisfaction.

Intimacy had a significant actor-effect on couple satisfaction. These results again confirm those from previous research [26,30,31,32] on how intimacy positively impacts couple satisfaction. However, only women’s level of intimacy significantly affected their partners’ couple satisfaction, which means that a higher level of intimacy on women’s part will impact in a positive way men’s level of couple satisfaction. Previous research tended to investigate different parts of intimacy, and the results depended on which ones were analyzed. For example, women’s and men’s responsiveness influenced each other’s couple satisfaction, while women’s disclosure influenced only their couple satisfaction [67]. On a similar note, when analyzing the effects of emotional and sexual intimacy, only the first one had a partner effect [30]. **Therefore, it is possible that general intimacy does not strongly influence partners as much as specific types of intimacy would**.

As for the impact of the use of interpersonal affect-improving strategies, in both models that contained it, a significant and positive effect was observable in women’s cases. Their usage of these strategies had a relevant impact on their couple satisfaction. The use of affect-worsening strategies was significant only when the model contained the commitment variable, and it showed that women’s employment of these strategies negatively impacted their partners’ couple satisfaction. These results partially fit those from previous research, showing a positive impact of affect-improving strategies [37,40] and a negative effect of affect-worsening strategies [41,42,43] on couple satisfaction. In addition, the way women provide social support can explain further these results. Women provided more supportive strategies [68] and were more sensitive to their partners’ emotional needs [69]. Furthermore, this is also expected from them to fulfill their gender role [70].

Regarding the interactions between commitment or intimacy and interpersonal affect-improving or -worsening strategies, some proved significant, while others did not. We observed that the model that contained commitment and couple satisfaction and interpersonal affect-improving strategies as the moderator role had the most interaction effects. The interaction of women’s commitment and use of affect-improving strategies positively impacted women’s and men’s couple satisfaction. Therefore, in this case, we observed that the positive impact of commitment [9,21,22] and affect-improving strategies [37,40] on couple satisfaction remained even when they interact. The higher women’s commitment and use of interpersonal affect-improving strategies were, the higher their and their partners’ couple satisfaction was.

A significant impact was observable on women’s and men’s couple satisfaction from the interaction of men’s commitment and women’s use of interpersonal affect-improving strategies. Interestingly, here we observe that when men had a lower level of commitment, women’s use of more affect-improving strategies helped them have a higher level of couple satisfaction. Therefore, in this case, IER seemed to work as a buffering element between low commitment and couple satisfaction. However, for men’s couple satisfaction, it seems better if they had a high level of commitment and their partners use less affect-improving strategies. **We could also observe a similar situation for women’s couple satisfaction; if their partners had a high level of commitment, there were better outcomes if they used less interpersonal affect-improving strategies**. These results might seem counterintuitive, as we expected the best outcomes to appear when there was a high level of commitment and more interpersonal affect-improving strategies were used. Nonetheless, a possible explanatory mechanism can be given by the overuse of social support. When it is too much, it can feel overwhelming, become a social strain [71,72], and influence the emotional state in a negative manner [73], which in turn could negatively affect couple satisfaction [74,75]. Therefore, in this case, if men perceive their partners as trying too much to improve their emotional state, it could harm their couple satisfaction. In women’s case, the possible explanation can come from the emotional labor that is considered a significant part of women’s duty in a relationship [76]. As such, if women see their partners as committed and feel they do not need to expend that much emotional effort in their relationship, it increases their couple satisfaction.

When it comes to men’s couple satisfaction and commitment, their using more interpersonal affect-improving strategies will produce the best outcomes. Here, we observed that the effects of the interaction went in a similar direction as the first one presented for this model. Therefore, we can see how couple satisfaction is influenced by commitment [9,21,22] and affect-improving strategies [37,40].

The other models in which we observed significative interaction effects were those with interpersonal affect-worsening strategies as the moderator. In the case of commitment, it seems that both women and men had higher levels of couple satisfaction when men had higher levels of commitment and their partners used more affect-worsening strategies. When the levels of commitment were low, using fewer affect-worsening strategies influenced couple satisfaction positively than when more of these strategies were used. Similar to the model containing commitment, in the one with intimacy, we observed a significant interaction between men’s lower level of intimacy and women’s use of more interpersonal affect-worsening strategies and its positive impact on women’s couple satisfaction. These results may seem counterintuitive, as we would have expected that exhibiting behavior meant to induce or exacerbate negative emotions would negatively influence other relational outcomes. However, the overall tone of those interactions influenced the result. Even if there were many interactions based on criticism between the partners, not all of them necessarily produced bad outcomes for the relationship. Non-hostile criticism seems to have a positive connection for both women’s and men’s couple satisfaction [77]. Similarly, ridicule can contribute to a lower level of couple satisfaction [78]. However, its effects are dependent on how the individual relates themself to them [42]. If the individual enjoys being laughed at, their couple’s satisfaction will not suffer, but if they hate being ridiculed, there will be negative consequences for their relationship outcomes.

The results of this study are important because they show how IER can interact with already established variables, such as commitment and intimacy, to affect couple satisfaction. To our knowledge, this is the first study to investigate how interpersonal affect-improving and -worsening strategies interact with commitment and intimacy to influence couple satisfaction from a dyadic perspective. Using an APIMoM model helped us present a more nuanced view of the relations between the study’s variables. Although the research had its advantages, some limitations need to be addressed. This study’s sample consisted of primarily young college students, and most couples were not cohabiting at the moment of data collection. Therefore, it would be interesting to conduct the same investigation on older participants in different stages of life with different responsibilities. Similarly, conducting this investigation on an all-cohabiting sample may provide a different perspective on the analyzed relations. In addition, causal studies (longitudinal or experimental) can add to the reliability of the results and therefore improve the research approach. Moreover, more of the participants’ background information (such as mental status, family of origin, education level, or financial status) was not addressed in the present study, which constitutes a limit, as it may have influenced the manner in which individuals use IER strategies and their couple satisfaction. Future research should consider and explore the impact of the abovementioned variables on the use of IER strategies and couple satisfaction.

## 5. Conclusions

This study aimed to explore the effects of commitment and intimacy on couple satisfaction and the moderator effect of interpersonal emotion regulation, in both its forms, as affect-improving and affect-worsening strategies. This research explored all the aforementioned effects on a younger population; as such, the following summary of the results and their implications in research and practice is primarily applicable to younger individuals.

As a plus of the present study, the models we computed (APIMoM) consider the partners as different units. Therefore we could analyze both the effects of their predictors on their outcomes and their partners’.

In the following paragraph we present the summary of this study’s results. The results highlight commitment and intimacy’s impact on couple satisfaction from both the actor’s and the partner’s perspectives. In all the models, commitment and intimacy significantly and positively impacted individuals’ couple satisfaction. The partner effect for commitment was significant only in the presence of interpersonal affect-worsening strategies. As for intimacy, only women’s level of intimacy appeared to affect men’s level of couple satisfaction. In addition, women’s use of interpersonal affect-improving strategies contributed to their couple satisfaction. When considering commitment, women’s use of interpersonal affect-worsening strategies decreased men’s couple satisfaction. Furthermore, this study showed how the interactions between commitment, intimacy, and IER strategies impacted couple satisfaction.

Therefore, these results shed some light on the previously unexplored moderating effect of interpersonal emotion regulation on the relations between commitment, intimacy, and couple satisfaction. As such, this research advances the knowledge of the ways interpersonal emotion regulation interacts with already established variables in the study of romantic relationships to affect couple satisfaction.

Consequently, these results can also offer some relevant information and hints for therapists working with couples. Practitioners can use these results to shape more accurate intervention plans depending on their initial assessment of the couple. As presented above, some of the results show that one of the partners can bring a more significant change to the overall couple satisfaction, and this can help the practitioners create a plan that will better fit the current couple they are working with rather than a general plan that will produce results only in some situations. For example, understanding that putting effort into consolidating the commitment aspect of the relationship produces good results when also working on the use of affect-improving IER strategies, but when developing the intimacy part of the relationship, it is not as important to work on the use of IER strategies, unless they are the affect-worsening type. Differentiating between what is relevant or not to spend time on in counseling or therapy can be extremely important, as it can represent the difference between building and continuing with a healthy relationship or separating because nothing changes.

## Figures and Tables

**Figure 1 ijerph-19-10506-f001:**
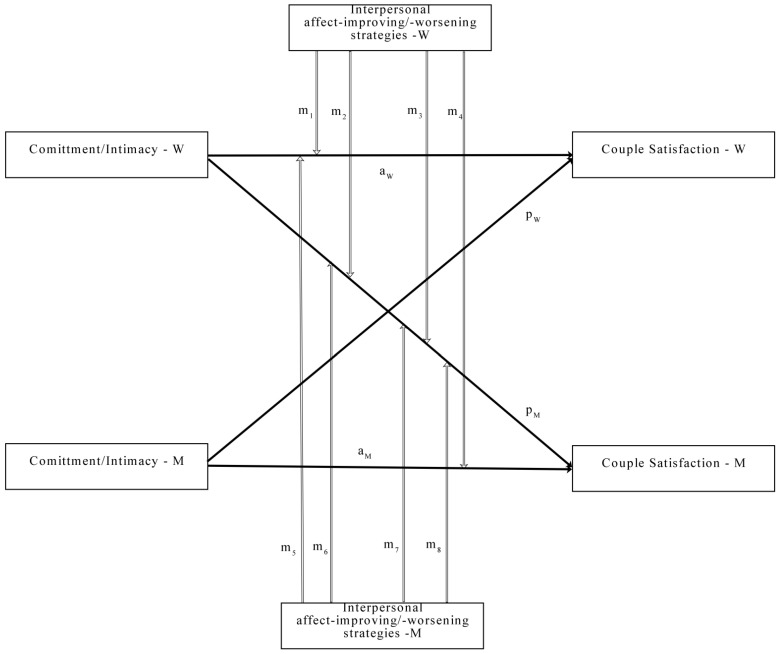
Conceptual moderated actor–partner interdependence model. Commitment/Intimacy—independent variable; Couple satisfaction—dependent variable; Interpersonal affect-improving/ affect-worsening strategies—moderator. a = actor effect; *p* = partner effect; m = moderator effect; W = women; M = men.

**Figure 2 ijerph-19-10506-f002:**
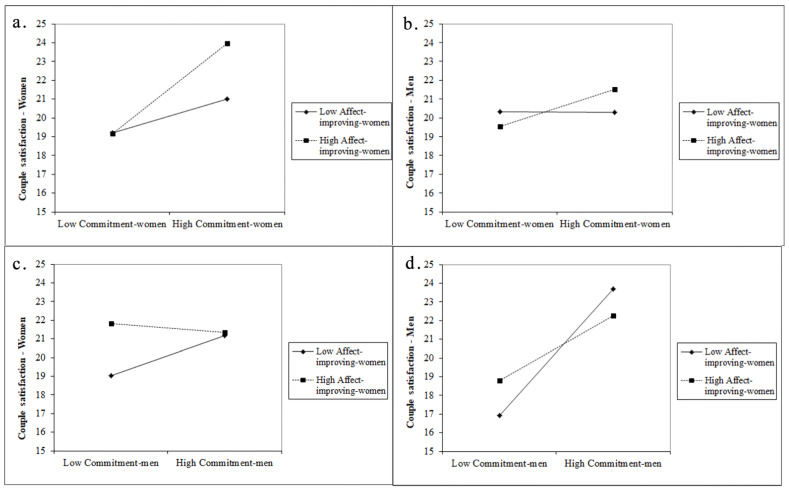
Women’s and men’s couple satisfaction with women’s or men’s low and high commitment for women that use less vs. more interpersonal affect-improving strategies: (**a**) women’s satisfaction with their commitment and use of interpersonal affect-improving strategies; (**b**) men’s satisfaction with women’s commitment and use of interpersonal affect-improving strategies; (**c**) women’s satisfaction with men’s commitment and women’s use of interpersonal affect-improving strategies; (**d**) men’s satisfaction with their commitment and women’s use of interpersonal affect-improving strategies.

**Figure 3 ijerph-19-10506-f003:**
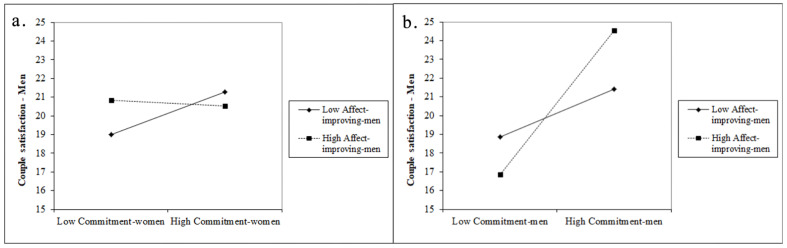
Men’s couple satisfaction with women’s or men’s low and high commitment for men that use less vs. more interpersonal affect-improving strategies: (**a**) men’s satisfaction with women’s commitment and men’s use of interpersonal affect-improving strategies; (**b**) men’s satisfaction with their commitment and use of interpersonal affect-improving strategies.

**Figure 4 ijerph-19-10506-f004:**
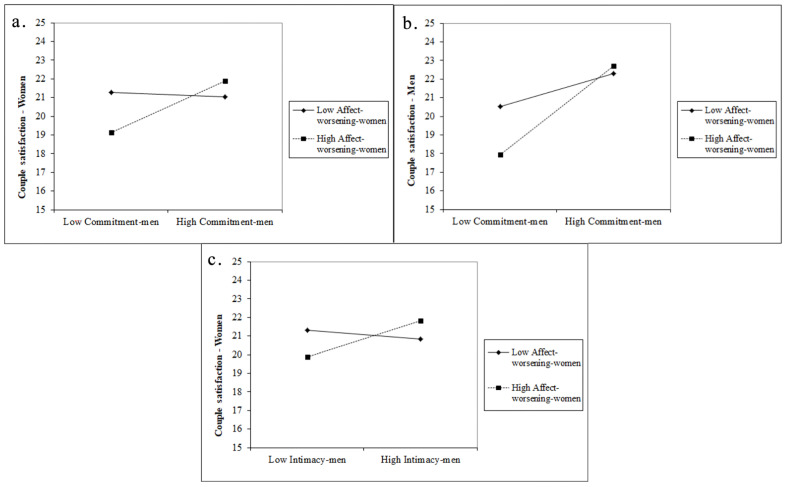
Women and men’s couple satisfaction with men’s low and high commitment (**a**,**b**) intimacy (**c**) for women that use less vs. more interpersonal affect-worsening strategies.

**Table 1 ijerph-19-10506-t001:** Descriptive statistics and paired-sample *t*-tests for this study’s variables.

	Mean	SD	Paired Sample Correlation	Mean Difference	SD (Mean Difference)	t	df
Commitment			0.509 **	1.18	8.58	1.58	130
Women	50.79	7.07					
Men	49.6	9.65					
Intimacy			0.559 **	1.4	15.13	1.06	130
Women	138.97	16.43					
Men	137.57	15.77					
Affect-improving strategies			0.425 **	−0.18	3.86	−0.52	130
Women	25.71	3.76					
Men	25.89	3.42					
Affect-worsening strategies			0.498 **	0.56	2.49	2.6 *	130
Women	5.51	2.6					
Men	4.95	2.35					
Couple Satisfaction			0.693 **	0.02	2.62	0.07	130
Women	20.84	3.22					
Men	20.82	3.45					

Note. ** *p* < 0.01, * *p* < 0.05.

**Table 2 ijerph-19-10506-t002:** Zero-order correlations for the main variables of the study by gender.

Variables	1	2	3	4	5	6	7	8	9
1. Commitment—W	1								
2. Commitment—M	0.488 **	1							
3. Affect-improving—W	0.248 *	0.067	1						
4. Affect-improving—M	0.226 *	0.377 **	0.395 **	1					
5. Affect-worsening—W	–0.338 **	–0.224 *	–0.187 *	−0.116	1				
6. Affect-worsening—M	–0.277 **	–0.315 **	−0.118	−0.116	0.499 **	1			
7. Intimacy—W	0.497 **	0.311 **	0.249 *	0.268 *	–0.503 **	–0.386 **	1		
8. Intimacy—M	0.358 **	0.512 **	0.12	0.370 **	–0.331 **	–0.369 **	0.557 **	1	
9. Satisfaction—W	0.618 **	0.440 **	0.359 **	0.318 **	–0.351 **	–0.300 **	0.565 **	0.421 **	1
10. Satisfaction—M	0.505 **	0.617 **	0.198 *	0.354 **	–0.348 **	–0.343 **	0.502 **	0.634 **	0.679 **

Note. ** *p* < 0.01, * *p* < 0.05; W = woman variable, M = man variable.

**Table 3 ijerph-19-10506-t003:** Model predicting couple satisfaction from commitment, moderated by women’s and men’s interpersonal affect-improving and affect-worsening strategies.

	Model with Affect-Improving IER Strategies				Model with Affect-Worsening IER Strategies			
Effect	Unstand. Coeff.	Stand. Coeff.	SE	*p*	Unstand. Coeff.	Stand. Coeff.	SE	*p*
Actor Effect of Commitment								
Woman	**0.234**	**0.515**	**0.036**	**<0.001**	**0.229**	**0.503**	**0.035**	**<0.001**
Man	**0.265**	**0.729**	**0.038**	**<0.001**	**0.17**	**0.475**	**0.027**	**<0.001**
Partner Effect of Commitment								
Woman	0.043	0.128	0.035	0.221	**0.066**	**0.199**	**0.025**	**0.009**
Man	0.07	0.144	0.038	0.067	**0.1**	**0.205**	**0.038**	**0.008**
Woman IER Strategies								
Woman	**0.197**	**0.233**	**0.061**	**0.001**	−0.123	−0.099	0.096	0.2
Man	0.028	0.031	0.066	0.669	**−0.211**	**−0.159**	**0.103**	**0.04**
Man IER Strategies								
Woman	0.087	0.094	0.069	0.204	−0.099	−0.072	0.111	0.373
Man	0.08	0.081	0.073	0.273	−0.078	−0.053	0.119	0.516
Woman Commitment by Woman IER Strategies								
Woman	**0.028**	**0.325**	**0.009**	**0.002**	−0.012	−0.068	0.015	0.392
Man	**0.019**	**0.209**	**0.01**	**0.049**	−0.017	−0.087	0.016	0.272
Man Commitment by Woman IER Strategies								
Woman	**−0.018**	**−0.245**	**0.007**	**0.012**	**0.03**	**0.214**	**0.012**	**0.015**
Man	**−0.023**	**−0.297**	**0.007**	**0.002**	**0.03**	**0.199**	**0.013**	**0.025**
Woman Commitment by Man IER Strategies								
Woman	0.01	0.084	0.012	0.437	−0.026	−0.165	0.015	0.073
Man	**−0.027**	**−0.22**	**0.013**	**0.04**	0	0.003	0.016	0.975
Man Commitment by Man IER Strategies								
Woman	−0.015	−0.161	0.011	0.166	0.002	0.015	0.013	0.889
Man	**0.039**	**0.401**	**0.011**	**<0.001**	−0.002	−0.012	0.014	0.908

Note. IER = Interpersonal Emotion Regulation. The significant effects are bold.

**Table 4 ijerph-19-10506-t004:** Model predicting couple satisfaction from intimacy, moderated by both women’s and men’s interpersonal affect-improving and affect-worsening strategies.

	Model with Affect-Improving IER Strategies				Model with Affect-Worsening IER Strategies			
Effect	Unstand. Coeff.	Stand. Coeff.	SE	*p*	Unstand. Coeff.	Stand. Coeff.	SE	*p*
Actor Effect of Intimacy								
Woman	**0.084**	**0.431**	**0.016**	**<0.001**	**0.091**	**0.465**	**0.018**	**<0.001**
Man	**0.103**	**0.471**	**0.017**	**<0.001**	**0.104**	**0.474**	**0.018**	**<0.001**
Partner Effect of Intimacy								
Woman	0.024	0.118	0.017	0.15	0.023	0.112	0.018	0.204
Man	**0.047**	**0.223**	**0.016**	**0.004**	**0.04**	**0.19**	**0.018**	**0.031**
Woman IER Strategies								
Woman	**0.18**	**0.21**	**0.065**	**0.006**	−0.043	−0.034	0.119	0.72
Man	0.03	0.032	0.067	0.661	−0.091	−0.068	0.121	0.452
Man IER Strategies								
Woman	0.042	0.045	0.076	0.577	−0.084	−0.061	0.137	0.54
Man	0.082	0.081	0.079	0.301	−0.067	−0.046	0.139	0.631
Woman Intimacy by Woman IER Strategies								
Woman	−0.009	−0.137	0.005	0.093	0.002	0.035	0.006	0.698
Man	−0.01	−0.147	0.005	0.063	−0.001	−0.009	0.006	0.912
Man Intimacy by Woman IER Strategies								
Woman	0.003	0.055	0.005	0.498	**0.015**	**0.181**	**0.007**	**0.035**
Man	−0.007	−0.111	0.005	0.157	0.002	0.025	0.007	0.756
Woman Intimacy by Man IER Strategies								
Woman	−0.008	−0.117	0.006	0.182	−0.013	−0.192	0.009	0.143
Man	0.001	0.008	0.006	0.923	−0.006	−0.079	0.009	0.525
Man Intimacy by Man IER Strategies								
Woman	−0.008	−0.133	0.006	0.135	0.005	0.063	0.01	0.633
Man	−0.004	−0.057	0.006	0.506	0.006	0.077	0.01	0.54

Note. IER = Interpersonal Emotion Regulation. The significant effects are bold.

## Data Availability

The data presented in this study are available on request from the corresponding author.

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
