# Peer review of "The Moderator Role of Interpersonal Emotion Regulation on the Associations between Commitment, Intimacy, and Couple Satisfaction"

_ijerph, 2022, doi:10.3390/ijerph191710506_

Round 1

Reviewer 1 Report

The article “Does it matter how we regulate each other’s emotions? The moderator role of interpersonal emotion regulation on the associations between commitment, intimacy, and couple satisfaction” investigated how commitment and intimacy influenced couple satisfaction and the moderator role of interpersonal emotion regulation (affect-improving and -worsening strategies) using a dyadic approach. Some novel and interesting results were obtained, but there were still many deficiencies as follows:

In the Materials and Methods

1. The participants were all so young, and their background information, eg. Education level, financial status, their families of origin, mental status and so on, these factors would have some part influence to their emotion regulation and setifaction.

2. The questionnaire used in the research should be completely displayed to readers through supplementary materials or other forms.

Language of the article

The full text should be carefully proofread, especially the tense used in the article.

1. In the part of introduction, the expression of previous research progress should be appropriate in the past tense.

2. page one, line 36, the word “Among” should be deleted.

3. page two, line 62, what is the word “wishto”? “wish to”?

4. page 7, line 236, “whouse” should be “who use”?

……

Author Response

Dear reviewer,

We would like to thank you for the thoughtful comments. We believe that they helped deliver a stronger paper. We updated the text according to the comments, the changes being highlighted using track changes.

  1. The participants were all so young, and their background information, eg. Education level, financial status, their families of origin, mental status and so on, these factors would have some part influence to their emotion regulation and setifaction.

We address this issue as a limit at the end of the Discussions section, on page 19, as we did not collect additional information about this study’s participants.

  1. The questionnaire used in the research should be completely displayed to readers through supplementary materials or other forms.

We added the questionnaires in the Appendix section of the article, starting with page 20.

Language of the article

The full text should be carefully proofread, especially the tense used in the article.

  1. In the part of introduction, the expression of previous research progress should be appropriate in the past tense.
  2. page one, line 36, the word “Among” should be deleted.
  3. page two, line 62, what is the word “wishto”? “wish to”?
  4. page 7, line 236, “whouse” should be “who use”?

……

We are grateful for the attention given to the details of the text and for helping us discover a significant problem with our manuscript that we did not observe before uploading it. There appears to be an incompatibility between the authors' versions of Microsoft Word, which has resulted in some of the words being merged. We remedied this problem. We also addressed the issue concerning the verbs' tenses.

Reviewer 2 Report

Congratulations to the authors for the manuscript quality. I will make just a few suggestions. My first one is to delete the question that starts the title once shorter titles make the message more accessible to the potential reader. The question is unnecessary to communicate the idea.

The second and third suggestions concern the results section. I recommend including an opening paragraph stating how the results will be presented. Many tables make it difficult to synthesize the results while reading, making it necessary to return to previous pages and the method section to consult the variables included in the models. Moreover, I recommend unifying the tables with the prediction models, as the tested effects are similar (for instance, tables 3 and 4). It is dispensable to separate the tables by strategy; the advantage is that it facilitates the reader's interpretation, comparing two strategies in the same table. The third suggestion is to present a synthesis of principle results in a table or figure for the effects of the variables on the women and men partners in the final results section, which means before the discussion section. It will prepare the readers for the discussion section. The fourth and final suggestion is to explore the practical implications of the empirical study. The authors mentioned at the end a short and superficial statement that the study offers hints for therapists working with couples. However, I think it would be helpful to provide more details on what kind of contribution the study offer to therapists or couples counselors. 

Author Response

Dear reviewer,

We would like to thank you for the thoughtful comments. We believe that they helped deliver a stronger paper. We updated the text according to the comments, the changes being highlighted using track changes.

 My first one is to delete the question that starts the title once shorter titles make the message more accessible to the potential reader. The question is unnecessary to communicate the idea.

As recommended, we deleted the first part of the original title.

I recommend including an opening paragraph stating how the results will be presented.

We introduced a paragraph at the beginning of the Results section, on page 6, describing how we present the results, the order, and how the models were constructed.

Moreover, I recommend unifying the tables with the prediction models, as the tested effects are similar (for instance, tables 3 and 4). It is dispensable to separate the tables by strategy; the advantage is that it facilitates the reader's interpretation, comparing two strategies in the same table. The third suggestion is to present a synthesis of principle results in a table or figure for the effects of the variables on the women and men partners in the final results section, which means before the discussion section. It will prepare the readers for the discussion section.

For a more precise presentation of the results, we united Tables 3 and 4, and Tables 5 and 6, considering the two moderators, and we also bolded the significant results for each model. We consider that a separate table or figures (we were unable to resume all the results in one single figure) would add bulk to the text and have a similar effect as the previous larger number of tables. Therefore, we consider that bolding the significant results is the safer approach.

The fourth and final suggestion is to explore the practical implications of the empirical study. The authors mentioned at the end a short and superficial statement that the study offers hints for therapists working with couples. However, I think it would be helpful to provide more details on what kind of contribution the study offer to therapists or couples counselors. 

We address this recommendation at the end of the Conclusions section on page 20.

Round 2

Reviewer 1 Report

The author has revised the manuscript in detail according to the review comments, but there are still some aspacts that need further improvement.

The results and conclusions are not accurate enough. The study subjects are  young people, not people of all ages.

And the conclusion are not concise and need to be further revised 

Author Response

Dear reviewer,

We would like to thank you once again for the thoughtful comments. They helped us to give more depth to the text. We updated the text according to the comments, the changes being highlighted again using track changes.

The results and conclusions are not accurate enough. The study subjects are  young people, not people of all ages.

And the conclusion are not concise and need to be further revised 

In the first line of the Participants section, we added the term “young” to describe the participating couples to introduce the idea that this study’s results mainly apply to a younger population. We also added a small paragraph in the Results section, before the preliminary analysis, in which we say that the participants are young and the results can not be reliably generalized to older individuals. In order to reiterate the idea that this study’s participants are young, we added a phrase at the end of the first paragraph from the Discussion section, which states this fact. Similarly, we added a phrase at the end of the first paragraph from the Conclusions section, stating that the results and their implications primarily apply to younger individuals. Additionally, we made some modifications to the Conclusions section in order to make it easier to read.